# The quality of life and well-being of nursing home residents and their relatives during the COVID-19 pandemic: A quantitative approach

**Philippe Schneider**[1], **Maryline Abt**[1], **Christine Cohen**[1], **Nadine Marmier**[2], **Lucia Ribeiro Carvalho**[2], **Claudia Ortoleva Bucher**[1]*

**1** La Source School of Nursing, University of Applied Sciences and Arts Western Switzerland (HES-SO), Lausanne, Switzerland, **2** Faculty of Biology and Medicine, Institute of Higher Education and Research in Health Care (IUFRS), University of Lausanne, Switzerland

\* c.ortolevabucher@ecolelasource.ch

## Abstract

### Background

During the COVID-19 pandemic, nursing homes implemented protection measures to limit contact with others. Measures implemented in Switzerland included in-room isolation, prohibitions on group eating, and limitations on visiting and group activities. Nursing home residents thus experienced long periods of social isolation, which could have impacted their well-being and that of their relatives (whether direct family or other loved ones). The present study aimed to quantitatively describe and compare the quality of life, well-being and coping strategies of nursing home residents and their relatives during the implementation of protection measures against COVID-19 in the French-speaking part of Switzerland.

### Method

136 residents and 47 relatives from 13 nursing homes responded to the Perceived Stress Scale (PSS), the Brief Coping Orientation to Problems Experienced (Brief-COPE) questionnaire, the Impact of Event Scale-6 (IES-6) and the Post-Trauma Growth Inventory (PTGI). The only (minimal) difference between groups was that resident's QoL was evaluated using the WHOQOL-OLD, and relatives' QoL was evaluated using the WHOQOL-BREF scale. Non-parametric T-test were used to compare between group when possible, and linear regression to evaluate predictor of quality of life.

### Results

Results showed that relatives coped better with lockdown isolation than did residents, residents developed lower levels of post-traumatic stress symptoms than relatives but there was no difference between the groups regarding perceived stress. Multiple linear regression calculations on indicators of quality of life showed that overall quality of life was most impacted by perceived stress. In the Psychological Health subscale, the familial link also reached significance: relatives who were part of a resident's direct family had a better psychological QoL than other loved ones from outside.

**Funding:** This study was financially supported by the University of Applied Sciences and Arts Western Switzerland (https://www.hes-so.ch) in the form of a grant for internal research projects (09-O20) received by COB. No additional external funding was received for this study. The funders had no role in study design, data collection and analysis, decision to publish, or preparation of the manuscript.

**Competing interests:** The authors have declared that no competing interests exist.

## Conclusion

The protection measures against COVID-19 isolated nursing residents, which impacted their quality of life and distanced residents and their relatives. In both populations, stress symptoms were the single most important predictor of quality of life.

## Introduction

Once identified in December 2019, the World Health Organization quickly reclassified the COVID-19 disease outbreak as a pandemic in March 2020. Older people were particularly at risk of developing fatal symptoms of the disease. Nursing homes quickly devised protection measures for their older adult residents. In Switzerland, these measures comprised in-room isolation, limitations on group activities, communal dining and visits, the use of personal protective equipment (PPE), particularly surgical masks, and maintaining distances [1]. Although these measures were taken in good faith and prioritised somatic health protection, there was a chance they could also have a detrimental impact on residents' quality of life (QoL) and well-being.

Residents' relatives have also been impacted by protection measures [2–4]. Relatives who usually invested time and energy in visiting residents were sometimes unable to see them for several weeks at a time, which could have diminished their QoL, too. Interestingly, few studies have jointly investigated the psychological well-being of residents and their relatives in the face of the measures implemented in nursing homes to prevent COVID-19 infection. A rapid review of studies evaluating the impact of protection measures on the well-being of nursing home residents and their relatives [5] showed ample evidence of harm to the psychological health of both groups. Only a minority of these articles investigated relatives, compared to nursing home residents.

The present study aimed to quantitatively describe the QoL and well-being of nursing home residents and their relatives (whether direct family or other loved ones) during the implementation of protection measures against COVID-19 in the French-speaking part of Switzerland. We also sought to assess potential coping mechanisms used by residents and relatives to cope with these measures. To this end, we recruited one sample of nursing home residents and another of their relatives (family members and other loved ones) to answer a series of questionnaires investigating their perceived stress, overall QoL, post-traumatic symptoms and post-trauma growth, and we compared results between those populations. This study was framed within the Neuman systems model [6], in which each person (or dyad) exposed to stressors has certain resources with which to maintain their health (well-being).

We aimed to measure several theoretical constructs. Information on the actual measurement scales used is in the Methods section. This study defines coping mechanisms as "thoughts and behaviors used to manage the internal and external demands of situations that are appraised as stressful" [7]. Perceived stress is defined as "the degree to which situations in one's life are appraised as stressful" [8]. This matches the transactional model of stress and coping [9], which stipulates that the degree to which an event is stressful depends, at least in part, on subjective perceptions of its stressfulness. QoL is defined as a person's perception of their position in life in the context of the culture and value system in which they live and in relation to their goals, expectations, standards and concerns [10]. Post-traumatic symptoms correspond to the symptomatology that can develop after living through traumatic events. The present study used a questionnaire measuring three domains of symptomatology that can develop

out of post-traumatic stress disorder: Intrusion, Avoidance and Hyperarousal. Finally, post-trauma growth is defined as "a positive psychological change experienced as a result of trauma or highly challenging situations" [11]. Posttraumatic Stress Disorder (PTSD) can manifest in various symptoms that do not necessarily meet the full diagnostic criteria. Indeed, individuals may experience distressing memories or flashbacks related to the trauma, albeit less frequent or intense than those seen in full PTSD cases) [12]. Avoiding reminders of the trauma, such as places or conversations, can occur, but may not be pervasive enough to meet full criteria [13] and symptoms such as irritability or heightened startle response can be observed, though they may not be as severe [14]. While these subthreshold symptoms can significantly impact an individual's quality of life, they may not be recognized as PTSD, leading to potential under-treatment.

When possible, we aimed to compare the scores between groups and analyse multiple linear regression equations to evaluate the impact of those scores on both groups' overall QoL, as measured by the WHOQOL scales. Comparison between groups would allow to investigate whether both group suffered from COVID-19 restrictions at the same level, or if they differed on how they lived the COVID-19 lockdown. Since both samples originate from the same socio-cultural background, difference between samples can be more directly imputed to difference in how the COVID-19 lockdown impacted their lives.

## Methods

The present study was part of a larger project that gathered quantitative and qualitative data from nursing home residents and their relatives on how the protection measures taken against COVID-19 affected their QoL and well-being. That project's protocol [15] and its qualitative part [16] have been published previously, and the present article describes its quantitative part.

This part of the study followed a questionnaire-based transversal correlational design. Both participant groups were recruited during the same period and answered the same questionnaire (apart from the WHOQOL questionnaire: see the Scales used and data gathering subsection). Participants were recruited and data were gathered between March 1 and August 31, 2021. This study was validated by the Human Research Ethics Committee of the Canton of Vaud (CER-VD, project ID 2020–02397).

### Population

Our sample consisted of 136 nursing home residents (106 women) from 13 nursing homes in the French-speaking part of Switzerland. Age of our participants ranged from 65 to 90+ years, with most participants being 80–89 years old. Approximately half of the nursing homes participating in the project were in urban areas, with the other half situated in the countryside. The number of beds in these nursing homes ranged from 48–350. In addition, 47 relatives (30 women; family members and other loved ones) of nursing home residents also took part in the investigation. Their aged ranged from 18 to 75, with most participants being in the 61–74 years old range. The participants in this study were a convenience sample. Advertisements for the study were posted in the nursing homes participating in the project. The inclusion criteria for residents were being 65 years old or more, living in a nursing home, having no major cognitive impairments, and understanding and speaking French. The inclusion criteria for relatives were understanding and speaking French, having no major cognitive impairments and having a relative or loved one living in a nursing home. All participants signed an informed consent form and were free to withdraw from the study at any moment. They also participated in the qualitative part of this project, which allowed to ensure that they had the cognitive capacity to respond to our series of questionnaire.

## Scales used and data gathering

Our participants responded to a series of self-administered questionnaires investigating their perceived stress, QoL, post-trauma growth and coping responses to stressful events such as a lockdown or a pandemic. Both samples responded to the Perceived Stress Scale (PSS, Cronbach's α = 0.81 [17], the Brief Coping Orientation to Problems Experienced (Brief-COPE, Cronbach's α = 0.71–0.82 [18] questionnaire, the Impact of Event Scale-6 (IES-6, Cronbach's α = 0.81–0.93 [19] and the Post-Trauma Growth Inventory (PTGI, Cronbach's α = 0.94 [20]. The only (minimal) difference between groups was that resident's QoL was evaluated using the WHOQOL-OLD (Cronbach's α = 0.72–0.83 [21], and relatives' QoL was evaluated using the WHOQOL-BREF scale (Cronbach's α = 0.59–0.74 [22]. Both samples responded to a different version of the WHOQOL questionnaire, because each questionnaire is better suited to measure each sample's well-being. All the scales were administered in their validated French-language versions. References and a short description of each scale are shown in Table 1. Participants also responded to a demographic questionnaire, collecting age and sex from both groups and the frequency of visits and the type of link (familial or not) for relatives. The experimenter was present in the room when participants from the resident sample answered the series of questionnaire, and they could ask the experimenter for clarification if needed. Raw data and codebook can be found on the following OSF page: https://osf.io/4n72v/.

## Data analysis plan

Participants' scores for each scale and subscale were calculated as described by the scales' authors. The mean scores on questionnaires completed by both groups (residents and

**Table 1. Study questionnaires and scales.**

| Scale Name (Authors names) | Description | Scale structure | Sample |
|---|---|---|---|
| Coping Orientation to Problems Experienced (Brief-COPE) [23] | Questionnaire designed to evaluate coping strategies. | Twelve subscales, each corresponding to a coping strategy: Active coping, Planification, Instrumental support, Emotional support, Expression of emotions, Positive reinterpretation, Acceptance, Denial, Blame, Humour, Religion, Distraction, Drug consumption and Disengagement. Higher scores mean greater use of this strategy. | Residents and Relatives |
| Impact of Event Scale–6 (IES-6) [19] | Questionnaire designed to screen for post-traumatic stress disorder. Each subscale represents a type of PTSD symptom. | One overall score and three subscales: Avoidance, Hyperarousal, and Intrusion. Higher scores mean higher levels of symptoms. | Residents and Relatives |
| Perceived Stress Scale (PSS) [24] | Questionnaire designed to evaluate perceived (felt) stress. | Single score, with a higher score meaning higher perceived stress. | Residents and Relatives |
| Post-traumatic Growth Inventory (PTGI) [11] | Questionnaire designed to evaluate post-trauma growth and self-improvement. | One overall score and five subscales: Appreciation of life, New Possibilities, Personal Strength, Relation to Others and Spiritual Change. Higher scores mean better post-traumatic growth. | Residents and Relatives |
| World Health Organization Quality of Life–OLD FR (WHOQOL-OLD-FR) [21] | Questionnaire designed to investigate QoL. The validated French-language version is designed specifically for French-speaking older adults. | One general score and six subscales: Autonomy (AUT), Death and End-of-Life (DAD), Intimacy (INT), Past, Present and Future activities (PPF), Sensorial Abilities (SAB) and Social Participation (SOP). Higher scores mean better QoL in that dimension. | Residents only |
| World Health Organization Quality of Life–BREF French Version (WHOQOL-BREF) [22] | questionnaire investigating QoL. | Four subscales: Physical Health, Psychological Health, Social Relationships and Environment. A higher score means better QoL in that dimension. | Relatives only |

List of the scales used in this research project (and descriptions of them). All the participants responded to all the questionnaires except for the WHOQOL-OLD, which was only completed by residents, and the WHOQOL-BREF, which was only completed by relatives.

relatives) were compared, and a t-test was used to measure potential differences between the groups. Values from these tests were corrected for multiple testing within each scale using a Bonferroni correction. We also analysed separate multiple linear regression equations for residents and relatives to evaluate which kinds of demographic variables (age, sex) and scale scores (PTGI, PSS and IES) predicted general and subscale scores for the WHOQOL-BREF (for relatives) and the WHOQOL-OLD (for residents). In addition, the frequency of relatives' visits and the social links between the relative and their resident (dichotomized as family member or not) were added as predictors in the relatives' sample analysis. Comparisons between the groups were made using the Wilcoxon–Mann–Whitney signed rank test (the wilcox.test() function), and all the linear regression analyses were performed using the lm() function from the Stat package in R software [25]. The alpha level was set at 0.05, and $p$-values from the analysis were corrected for multiple comparisons using a Bonferroni correction. The Wilcox–Mann–Whitney test was used because the results did not meet the requirements for parametric statistical tests (normal distribution).

## Results

Mean score and p-values of the Wilcoxon-Mann-Whitney test can be found in Table 2.

### Perceived stress scale

We found no statistically significant difference between residents' and relatives' mean PSS scores. Overall scores were similar to what has been found in other samples [24].

### Brief coping orientation to problems experienced

Answers to the Brief-COPE questionnaire were sorted into three categories, as Dias *et al.* [26] suggested. We assessed each participant's score in each type of coping strategy and compared relatives' and residents' scores separately for each type of coping strategy using t-tests. We found no statistically significant differences between residents and relatives regarding their Avoidant Coping ($p$ = 0.95, 95% CI [-0.14; 0.09]) and Problem-Focused Coping ($p$ = 0.11, 95% CI [-0.02; 0.26]) strategies. However, relatives had significantly higher scores than residents regarding the Emotion-Focused Coping strategy ($p < 0.001$, 95% CI [0.08; 0.31]).

### Impact of event Scale-6

We compared residents' and relatives' scores on the IES-6, revealing that relatives scored higher on every subscale and overall. Relatives had a higher overall score ($p < 0.001$, 95% CI [1.8; 4.6]), Avoidance score ($p < 0.05$, 95% CI [0.01; 0.5]), Hyperarousal score ($p < 0.001$, 95% CI [0.19; 0.75]) and Intrusion score ($p < 0.001$, 95% CI [0.51; 1.26]) than residents.

### Post-trauma growth inventory

We compared relatives' and residents' overall PTGI and subscale scores. Independent t-tests were performed on each score to test whether residents' and relatives' mean scores differed. A Bonferroni correction was applied to their $p$-values to correct for multiple tests. Relatives had higher scores than residents on the New Possibilities (p < 0.001, 95% CI [0.8; 2.6]) and Personal Strength ($p < 0.001$, 95% CI [0.9; 2.8]) subscales. The differences between residents and relatives were not significant for the overall score ($p$ = 0.06, 95% CI [-0.1; 9.1]) or the other subscales: Appreciation of Life ($p$ = 0.24, 95% CI [-0.2; 1.8], Relation to Others ($p$ = 0.98, 95% CI [-1.2; 0.9]) and Spiritual Change: ($p$ = 0.18, 95% CI [-0.1; 2.0]),

**Table 2. Mean group scores for each scale and subscale.**

| Scale | Subscales (if any) | Group | | Difference (*p*-value) |
|---|---|---|---|---|
| | | Residents | Relatives | |
| PSS | | 19.8 (9.0) | 18.6 (7.8) | 0.6 |
| Brief-COPE | Avoidant Coping | 2.2 (0.3) | 2.1 (0.3) | 0.95 |
| | Emotion-Focused Coping | 1.9 (0.4) | 2.1 (0.3) | **< 0.001***** |
| | Problem-Focused Coping | 2.2 (0.5) | 2.3 (0.3) | 0.11 |
| IES-6 | Overall score | 4.8 (4.6) | 8.2 (4.0) | **< 0.001***** |
| | Avoidance | 0.6 (0.8) | 0.8 (0.8) | **< 0.05*** |
| | Hyperarousal | 0.6 (0.9) | 1.1 (0.8) | **< 0.001***** |
| | Intrusion | 1.2 (1.2) | 2.1 (1.1) | **< 0.001***** |
| PTGI | Overall score | 11.8 (10.8) | 16.9 (12.2) | 0.06 |
| | Appreciation of Life | 3.1 (3.2) | 3.9 (2.8) | 0.24 |
| | New Possibilities | 1.5 (2.4) | 3.2 (2.7) | **< 0.001***** |
| | Personal Strength | 1.6 (2.4) | 3.4 (3.0) | **< 0.001***** |
| | Relation to Others | 4.1 (3.7) | 4.0 (3.0) | 0.98 |
| | Spiritual Change | 1.5 (2.9) | 2.4 (3.1) | 0.18 |
| WHOQOL-OLD (Residents only) | Autonomy | 13.6 (3.0) | | |
| | Death and End-of-Life | 16.9 (3.3) | | |
| | Intimacy | 14.4 (3.7) | | |
| | Past, Present and Future activities | 14.1 (3.2) | | |
| | Sensory Abilities | 14.8 (4.6) | | |
| | Social Participation | 11.4 (2.7) | | |
| WHOQOL-BREF (Relatives only) | Environment | | 17.0 (2.9) | |
| | Physical Health | | 16.2 (3.1) | |
| | Psychological Health | | 15.3 (3.0) | |
| | Social Relationships | | 15.2 (3.1) | |

Mean scores (and standard deviations) for the different questionnaires completed by the resident and relative groups. The 'difference' column shows the p-value of the Wilcoxon-Mann-Whitney test used for the differences between groups. The alpha level for significance was set at 0.05, but p-values were corrected using a Bonferroni correction.

## Statistical predictors of quality of life

We used multiple linear regression analysis to evaluate which predictors included in the WHOQOL-OLD and WHOQOL-BREF questionnaires impacted residents' and relatives' QoL, respectively. We first calculated each participant's scores for each subscale and then used these as dependent variables in a multiple linear regression model, where mean PSS, PTGI and IES-6 scores and demographic variables (sex and age group for the residents' group; sex, age, frequency of visits and whether the relative was a family member or another loved one for the relatives' group) were the dependent variables. Results from the residents' group analysis are shown in Table 3, and results for relatives are shown in Table 4.

This analysis showed that PSS scores predicted overall WHOQOL-OLD scores and four subscale scores but not the Death and End-of-Life or the Intimacy subscales. The Death and End-of-Life subscale was the only model that did not reach significance.

The results for relatives showed similar patterns to those for residents. PSS scores predicted all four subscale scores; however, the model's *p*-values only reached the level of statistical significance in three of the four subscales. Higher scores in PSS induced lower scores in the Psychological, Social Relationships and Environment subscales. In the Psychological Health

**Table 3. Predictors of residents' QoL.**

| Outcomes of interest | PSS | IES | PTGI | Sex | Age | Model *P*-value | Adjusted R$^2$ |
|---|---|---|---|---|---|---|---|
| Overall Score | **4.1$^E$-6***** (**-0.48**) | 0.12 (0.16) | 0.31 (0.10) | 0.47 (-0.06) | 0.74 (-0.03) | **4.7$^E$-4***** | 0.15 |
| SAB | **4.0$^E$-3**** (**-0.29**) | 0.31 (0.10) | 0.18 (0.12) | 0.63 (-0.04) | **0.01*** (**-0.24**) | **8.2$^E$-3**** | 0.09 |
| AUT | **2.5$^E$-7***** (**-0.51**) | **8.0$^E$-3**** (**0.26**) | 0.99 (0.00) | 0.50 (0.06) | 0.81 (0.02) | **3.8$^E$-5***** | 0.18 |
| PPF | **1.0$^E$-4***** (**-0.39**) | 0.61 (0.05) | 0.25 (0.11) | 0.47 (-0.08) | 0.85 (-0.01) | **4.3$^E$-3**** | 0.10 |
| SOP | **2.6$^E$-5***** (**-0.41**) | 0.28 (0.10) | 0.11 (0.14) | 0.08 (-0.15) | 0.34 (-0.08) | **4.9$^E$-4***** | 0.14 |
| DAD | 0.21 (-0.12) | 0.49 (0.07) | **0.04*** (**-0.20**) | 0.46 (0.07) | 0.29 (0.10) | 0.12 | 0.03 |
| INT | 0.14 (-0.39) | 0.79 (0.05) | 0.09 (0.11) | 0.06 (-0.8) | 0.13 (-0.01) | **0.02*** | 0.07 |

*Outcomes of interest (overall score or subscale scores for the WHOQOL-OLD). Columns from PSS to Age give a p-value (standardised beta) for the predictor. Model p-value is the p-value for the whole model. Adjusted R$^2$ is the adjusted R-squared for the whole model. Predictors with a p-value < 0.05 are in bold. P-values*

* < 0.05

** < 0.01

*** < 0.001. SAB: Sensory Abilities; AUT: Autonomy; PPF: Past, Present and Future Activities; SOP: Social Participation; DAD: Death and End-of-Life; INT: Intimacy.

subscale, the familial link also reached significance. Relatives who were part of a resident's direct family had a better psychological QoL than other loved ones from outside it. The model for the Physical Health subscale was not significant.

## Discussion

This study aimed to document the results of various self-reported psychometric measurement instruments completed by a sample of 136 nursing home residents and 47 of their relatives. They answered a series of questionnaires aimed at evaluating their perceived stress (PSS), whether the 2020 COVID-19 lockdown resulted in post-traumatic stress symptoms (IES-6), their potential post-traumatic growth (PTGI) and their QoL (WHOQOL-OLD for residents

**Table 4. Predictors of relatives' QoL.**

| Outcomes of interest | PSS | IES | PTGI | Sex | Age | Visit Freq | Familial Link | Model *P*-value | Adjusted R$^2$ |
|---|---|---|---|---|---|---|---|---|---|
| Physical Health | **0.02*** (**-0.48**) | 0.11 (0.29) | 0.39 (-0.16) | 0.31 (0.17) | 0.51 (-0.12) | 0.67 (0.06) | 0.12 (-0.23) | 0.06 | 0.17 |
| Psychological Health | **0.02*** (**-0.46**) | 0.31 (0.18) | 0.40 (0.14) | 0.61 (-0.08) | 0.41 (0.14) | 0.31 (0.15) | **0.02*** (**-0.35**) | **0.02*** | 0.22 |
| Social Relationships | **0.009**** (**-0.51**) | 0.94 (-0.01) | 0.39 (0.15) | 0.09 (-0.28) | 0.41 (0.15) | 0.57 (0.08) | 0.30 (-0.15) | **0.049*** | 0.17 |
| Environment | **0.003**** (**-0.56**) | 0.44 (0.13) | 0.99 (0.00) | 0.16 (-0.22) | 0.64 (-0.08) | 0.18 (0.19) | 0.39 (-0.12) | **0.02*** | 0.23 |

*Outcomes of interest (overall score or subscale scores for the WHOQOL-OLD). Columns from PSS to Familial Link give a p-value (standardised beta) for the predictor. Model p-value is the p-value for the whole model. Adjusted R$^2$ is the adjusted R-squared for the whole model. Predictors with a p-value < 0.05 are in bold. P-values*

* < 0.05

** < 0.01

*** < 0.001.

and WHOQOL-BREF for relatives). We subsequently used statistical analysis to evaluate whether there were differences between residents' and relatives' responses to the scales and whether these scales impacted QoL.

## Differences between residents and relatives

Analysis of the Brief-COPE questionnaire showed that relatives and residents only differed in their emotion-focused strategy, which relatives used more than residents. The groups had similar scores for other coping strategies. This could be due to residents having less access to social contact during the lockdown than their relatives outside the nursing homes. Indeed, nursing home residents reported a heightened levels of loneliness during the COVID-19 pandemic, and thus less possibility to engage in emotion-focused strategies [1,16].

Results from the IES-6 showed that residents developed lower levels of post-traumatic stress symptoms than their relatives in every subscale and on the overall score. Residents developed less symptoms of Intrusion, Hyperarousal and Avoidance than their relatives, but both groups' overall scores were relatively low compared to the cut-off scores proposed by the IES-6's developers [27]. This is likely because the IES-6 was developed to evaluate the impact of a single specific event, whereas the lockdown lasted several weeks in 2020.

There was no significant difference between residents' and relatives' mean PSS scores, suggesting no differences in their stress symptomatology, and their overall PSS scores were similar to those found in other samples during the COVID-19 lockdown [28]. Interestingly, those other samples found that older people had lower levels of stress symptoms than younger people, but this was not the case in our sample.

Finally, residents' and relatives' PTGI scores only differed significantly in two subscales (New Possibilities and Personal Strength). Residents had a lower mean New Possibilities subscale score than relatives, likely because residents were often confined to their nursing homes, whereas relatives had more possibilities to do things outside. Interestingly, overall PTGI scores were relatively low, perhaps because data was gathered while some protection measures were still in place in nursing homes and still affecting the general population (although less stringently than at the beginning of the pandemic). This could have limited the extent to which nursing home residents and their relatives benefitted from post-traumatic growth. It may also be that, although protection measures had an impact on residents' and relatives' well-being, a prolonged lockdown was too dissimilar to a one-off traumatic event for which the PTGI was developed [29,30]. Furthermore, data were gathered more than a year after the beginning of the first COVID-19-related lockdown. It may be that the actual post-traumatic growth resulting from the COVID-19 lockdown was no longer evident in our samples.

## Quality of life predictor

Our nursing home residents' PSS scores negatively impacted their overall QoL scores and those of several QoL subscales (SAB, AUT, PPF and SOP). Indeed, the PSS score was the only predictive factor for these subscales, except for the AUT subscale, for which the IES score was also a significant predictor. These findings were congruent with the idea that lockdown measures mainly impacted residents' perceived stress. The AUT subscale was also modulated by the IES score, which is congruent with the fact that lockdown measures mainly involved limitations on residents' freedom of movement. It is important to note that all the models presented in this analysis had modest adjusted $R^2$ values, which showed that overall QoL was also based upon other factors not considered in our statistical models.

We found similar patterns of results with our sample of relatives. The PSS score was negatively associated with all the subscale scores and was the sole predictor in three out of four of

them. In addition, the resident's familial link was a predictive factor for the Psychological Health subscale score. The only model that did not reach a level of significance was the one for Physical Health, which seemed not to be affected by any of our predictors. This pattern of results showed that the PSS was again the main negative factor impacting QoL among our sample of relatives.

Patterns of results showed that the COVID-19 pandemic induced stress in both residents and relatives, which, in turn, impacted their QoL. This was in line with the results of other studies on the impact of protection measures against COVID-19 in nursing homes (see [5], for a review), with most of the literature on this subject showing their detrimental effects on the QoL of residents and their relatives. The qualitative results of this project [14] showed that residents reported being separated from their close-ones was one of the main complain they reported.

Although other factors, such a length of stay, could have also impacted resident's wellbeing, such a factor was not evaluated because the median total stay duration for resident in a nursing home in Switzerland is around 3 years, with minimal variance. Thus, adding this factor would have only limited interest in this specific context. Future studies should aim at investigating other factors that could impact residents wellbeing in nursing homes.

## Recommendation

Based on the quantitative results as well as the qualitative results from the sister article, several recommendations can be made to support nursing home residents and their relative's wellbeing during a pandemic such as the COVID-19 one. Three main recommendations were reported directly by residents themselves in the qualitative part of this project [16]: First, nursing home should ensure that relatives are allowed in the nursing home, although with personal protections (masks, gels,, etc.). This would alleviate resident isolation. Nursing home should also not stop all activity and keep a certain level of "normalness". Finally, a resident council can be created, so residents have the opportunity to voice their concerns and be more included in the actual decisions about themselves and their health. In 2023, on behalf of the Swiss Federal Office of Public Health, a panel of experts issued 7 postulates and 21 recommendations based on available scientific data and the testimonies of residents and their relatives, in order to better balance measures to protect life and maintain quality of life for a future pandemic. Several recommendations stress the need to maintain residents' self-determination and social contacts. The facility should also maintain life within the facility and provide effective communication both inside and outside the facility [1].

## Limitations

The present study had some limitations. First, the analyses presented were exploratory in design, meaning that the risk of a type 1 error in our statistical testing could be relatively high, especially for $p$-values close to the 0.05 threshold. Second, data were gathered one year after the beginning of the COVID-19 pandemic, and by this time, protection measures were less strict than during the first lockdown period (March–May 2020). People thus recalled their experience retrospectively, which could potentially generate some recall bias. Finally, since such data were not gathered routinely for both residents and relatives population pre-COVID-19, it is difficult to compare our results with a comparable sample pre-COVID-19.

## Conclusion

The protection measures taken against COVID-19 put significant restrictions on the movements of nursing home residents. These restrictions induced heightened levels of stress, which

in turn reduced residents' quality of life (QoL). However, they also impacted residents' relatives, who could not see their loved ones as often or in the same way as before the pandemic. Perceived stress was the single biggest predictor of a reduction in QoL. This is congruent with the idea that protection measures, although important to protect physical health, had a detrimental impact on the QoL and well-being of nursing home residents and their relatives. In case of a future pandemic similar to the COVID-19 pandemic, special attention should be paid to designing protection measures, especially for nursing homes, that take QoL into account.

## Author Contributions

**Conceptualization:** Maryline Abt, Christine Cohen, Claudia Ortoleva Bucher.

**Formal analysis:** Philippe Schneider.

**Funding acquisition:** Claudia Ortoleva Bucher.

**Investigation:** Maryline Abt, Christine Cohen, Nadine Marmier, Lucia Ribeiro Carvalho, Claudia Ortoleva Bucher.

**Methodology:** Claudia Ortoleva Bucher.

**Project administration:** Claudia Ortoleva Bucher.

**Writing – original draft:** Philippe Schneider.

**Writing – review & editing:** Philippe Schneider, Maryline Abt, Christine Cohen, Nadine Marmier, Lucia Ribeiro Carvalho, Claudia Ortoleva Bucher.

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
