## [Decision Letter · Decision Letter 0]

8 Oct 2024

PMEN-D-24-00288

The quality of life and well-being of nursing home residents and their relatives during the COVID-19 pandemic: a quantitative approach

PLOS Mental Health

Dear Dr. Philippe Schneider,

Thank you for submitting your manuscript to PLOS Mental Health. After careful consideration, we feel that it has merit but does not fully meet PLOS Mental Health’s publication criteria as it currently stands. Therefore, we invite you to submit a revised version of the manuscript that addresses the points raised during the review process.

The reviewers provide important insights for improving your paper. 

We look forward to receiving your revised manuscript.

Kind regards,

Martin Mabunda Baluku, Ph.D.

Academic Editor

PLOS Mental Health

Journal Requirements:

1. We ask that a manuscript source file is provided at Revision. Please upload your manuscript file as a .doc, .docx, .rtf or .tex.

Additional Editor Comments (if provided):

Reviewers' comments:

Reviewer's Responses to Questions

**Comments to the Author**

1. Does this manuscript meet PLOS Mental Health’s publication criteria? Is the manuscript technically sound, and do the data support the conclusions? The manuscript must describe methodologically and ethically rigorous research with conclusions that are appropriately drawn based on the data presented.

Reviewer #1: Yes

Reviewer #2: Partly

Reviewer #3: Yes

2. Has the statistical analysis been performed appropriately and rigorously?

Reviewer #1: Yes

Reviewer #2: Yes

Reviewer #3: Yes

3. Have the authors made all data underlying the findings in their manuscript fully available (please refer to the Data Availability Statement at the start of the manuscript PDF file)?

Reviewer #1: Yes

Reviewer #2: Yes

Reviewer #3: Yes

4. Is the manuscript presented in an intelligible fashion and written in standard English?

Reviewer #1: Yes

Reviewer #2: No

Reviewer #3: Yes

5. Review Comments to the Author

Reviewer #1: The authors report a study in which questionnaires were used to investigate the perceived stress and overall quality of life of residents in nursing homes and their relatives in the setting of the COVID-19 pandemic and resulting protection measures in Switzerland. They found no significant differences between the perceived stress of the residents and their relatives, and that stress symptoms were the most important predictor of quality of life. While an interesting paper, some points below could use more focus:

One of the aims discussed in the introduction is assessing coping mechanisms to cope with COVID-19 measures, but the mechanisms are not much discussed in the results or discussion.

The introduction covers that both residents and their relatives are impacted by protection measures. Why would a comparison between the two groups be necessarily needed?

Both samples of residents and their relatives, especially the relatives, are heterogeneous groups that would benefit from a more nuanced interpretation of results. For example, if relatives were found to have a higher hyperarousal score than residents, what do these results mean and how can this finding benefit the literature? The current analyses may be too general for the results to be truly meaningful. If possible, it would be interesting to compare each group before and after the COVID-19 measures were implemented or to include how long each resident has been in the nursing home to see if perhaps the length of stay may influence their coping or quality of life.

The finding that residents’ PSS scores negatively impacted their QoL scores is interesting but also somehow logically expected. The authors say that these were congruent with the idea that lockdown measures mainly impacted residents’ perceived stress. How were residents' perceptions of lockdown measures incorporated into the questionnaire? In turn, were other aspects that heavily influence QoL (e.g., physical health, culture, etc.) measured? If not, what are some potential directions for future studies?

An additional limitation is the fact that the sample came from one nursing home. More description of the nursing home (how big it is, whether it is located in the urban or countryside, and what specific measures were implemented and remained after 1 year) would be a good addition.

The paper would benefit from some suggestions on how to design new protection measures that take QoL into account. Some added literature into the discussion on what interventions have been proposed or implemented to help alleviate perceived stress from a pandemic and its following lockdown measures would help the paper be more fruitful.

Reviewer #2: PMEN-D-24-00288

The quality of life and well-being of nursing home residents and their relatives during the COVID-19 pandemic: a quantitative approach

- My appreciation for the opportunity to review the manuscript. I believe the manuscript addresses an important aspect of psychological wellbeing in light of the COVID-19 pandemic restrictions. I hope you find the comments below helpful.

- I believe that dually investigating the impact of COVID-19 measures on the psychological well-being of nursing home residents and their relatives may offer unique and valuable insights in the balance between physical and mental health. However, there are a few concerns:

- It is important that authors include some discussion of the existing literature on the relationship between isolation and wellbeing or quality of life. To say that “…there was a chance they (measures against COVID-19) could have a detrimental impact on residents’ quality of life” is to ignore any reference to existing literature.

- P. 5: “The present study used a questionnaire measuring three domains of symptomatology that can develop out of post-traumatic stress disorder: Intrusion, Avoidance and Hyperarousal.” It is my understanding that these domains do not develop out of PTSD, but are central to the diagnosis itself. So, the statement needs to be clarified. The authors should explain the rationale for choosing measures that only focus on Intrusion, Avoidance and Hyperarousal symptom clusters. The DSM-5 diagnostic criteria for PTSD includes many more components that these. It is important for the authors to provide a rationale for their decision.

- With regard to instruments used, it is important to provide the psychometric properties of any measures used. All measures used to gather quantitative data appear to be standardized. The authors should, therefore, provide relevant psychometric properties/information (reliability and validity) related to each of the measures.

- p. 10: I find this statement a bit confusing: “The mean resident and relative scores for the questionnaires and scales that both groups responded to were compared, and the t-test was used to quantify potential differences between the samples.” Perhaps, it may be written thus: The mean scores on questionnaires completed by both groups (residents and relatives) were compared, and a t-test was used to measure potential differences between the groups.

- p. 10: I do not see a need to include any description of the General Health Questionnaire (GHQ), since it was not used for any reported data in this study/manuscript.

- The data analysis section needs some more clarifying description. Notably, it is important for the authors to explain what “requirements for parametric statistical tests” the data did not meet? Additionally, it would be helpful for the reader to know some of the key assumptions of parametric statistical tests as these may provide insight into the authors’ decision-making and rationale. In other words, which of the three key assumptions (normal distribution, homogeneity of variance, and independence of the variable) did the sample data not meet?

- The discussion section needs to be contextualized within existing literature. As it stands, lacks details of how the results (significant or not) may be interpreted/understood within the context of current or extant literature.

Reviewer #3: ABSTRACT

In the Methods section of the abstract, specify the sample and the type of questionnaire used in this study. The results should report the key findings of the study based on the study objectives.

INTRODUCTION

The introduction is well-written. However, I have few comments for your consideration.

‘Interestingly, few studies have jointly investigated the psychological well-being of residents and their relatives in the face of the measures implemented in nursing homes to prevent COVID-19 infection.’

Can you provide references for some of the studies?

METHODS

‘The age of our participants ranged from 65 to 90+ years, with most participants being 80-89 years old’

How did you establish the mental capacity of the population to respond to the questionnaire?

“Our participants responded to a series of self-administered questionnaires investigating their own responses. perceived stress, quality of life, post-trauma growth, and coping responses to stressful events such as a lockdown or a pandemic”

Explain if the participants required assistance in responding to the questionnaires.

‘The only (minimal) difference between the groups was that resident’s quality of life was evaluated using WHOQOL-OLD, and relatives’ QoL was evaluated using the WHOQOL-BREF scale.’

Justify the use of different versions of the scale in this study.

DISCUSSION

The discussion can be improved by comparing the findings of this study with those of previous literature.

6. PLOS authors have the option to publish the peer review history of their article (what does this mean?). If published, this will include your full peer review and any attached files.

**Do you want your identity to be public for this peer review?** For information about this choice, including consent withdrawal, please see our Privacy Policy.

Reviewer #1: No

Reviewer #2: No

Reviewer #3: No

---

## [Decision Letter · Decision Letter 1]

26 Dec 2024

The quality of life and well-being of nursing home residents and their relatives during the COVID-19 pandemic: a quantitative approach

PMEN-D-24-00288R1

Dear Dr. Bucher, 

We are pleased to inform you that your manuscript 'The quality of life and well-being of nursing home residents and their relatives during the COVID-19 pandemic: a quantitative approach' has been provisionally accepted for publication in PLOS Mental Health.

Before your manuscript can be formally accepted you will need to complete some formatting changes, which you will receive in a follow up email. A member of our team will be in touch with a set of requests. Alongside these formatting changes, kindly address the two minor issues raised by the reviewers. 

Best regards,

Martin Mabunda Baluku, Ph.D.

Academic Editor

PLOS Mental Health

Reviewer Comments (if any, and for reference):

Reviewer's Responses to Questions

**Comments to the Author**

1. If the authors have adequately addressed your comments raised in a previous round of review and you feel that this manuscript is now acceptable for publication, you may indicate that here to bypass the “Comments to the Author” section, enter your conflict of interest statement in the “Confidential to Editor” section, and submit your "Accept" recommendation.

Reviewer #1: All comments have been addressed

Reviewer #3: All comments have been addressed

2. Does this manuscript meet PLOS Mental Health’s publication criteria? Is the manuscript technically sound, and do the data support the conclusions? The manuscript must describe methodologically and ethically rigorous research with conclusions that are appropriately drawn based on the data presented.

Reviewer #1: Yes

Reviewer #3: Yes

3. Has the statistical analysis been performed appropriately and rigorously?

Reviewer #1: Yes

Reviewer #3: Yes

4. Have the authors made all data underlying the findings in their manuscript fully available (please refer to the Data Availability Statement at the start of the manuscript PDF file)?

Reviewer #1: Yes

Reviewer #3: Yes

5. Is the manuscript presented in an intelligible fashion and written in standard English?

Reviewer #1: Yes

Reviewer #3: Yes

6. Review Comments to the Author

Reviewer #1: The aim of the paper brings light to the relationship between COVID-19 lockdown measures with the perceived stress of nursing home residents and their relatives and highlights the needs of underserved populations. The paper benefits from the various psychometric measurement instruments used.

The added paragraph of recommendations helps bring together readers’ understanding of the overall context and findings of the paper. There are minor errors throughout the manuscript (e.g., an additional/lack of space(s) after words, “aged” instead of ages in pg. 7) that would benefit from a thorough check over prior to publication.

Reviewer #3: Background

‘Older people were particularly at risk of developing fatal symptoms of the disease.’ Add reference to this statement.

“Interestingly, few studies have jointly investigated the psychological well-being of residents and their relatives in the face of the measures implemented in nursing homes to prevent COVID-19 infection.” List some of the studies conducted in this area.

You must justify the relevance of the study in the background. The gap this study will fill is not clearly stated in the background.

Methodology

‘Both samples responded to a different version of the WHOQOL questionnaire, because each questionnaire is better suited to measure each sample’s well-being.’ Attach a reference to this statement.

Discussion

It is suggested that the findings are compared with similar studies to draw similarities or other similarities of the findings of this study.

7. PLOS authors have the option to publish the peer review history of their article (what does this mean?). If published, this will include your full peer review and any attached files.

**Do you want your identity to be public for this peer review?** For information about this choice, including consent withdrawal, please see our Privacy Policy.

Reviewer #1: No

Reviewer #3: **Yes: **Sampson Opoku Agyemang
